# Multistate structures in a hydrogen-bonded polycatenation non-covalent organic framework with diverse resistive switching behaviors

Shimin Chen[1,3], Yan Ju[1,3], Yisi Yang [1], Fahui Xiang[1], Zizhu Yao[1], Hao Zhang[1], Yunbin Li[1], Yongfan Zhang[2], Shengchang Xiang [1], Banglin Chen [1] & Zhangjing Zhang [1] ✉

The inherent structural flexibility and reversibility of non-covalent organic frameworks have enabled them to exhibit switchable multistate structures under external stimuli, providing great potential in the field of resistive switching (RS), but not well explored yet. Herein, we report the 0D+1D hydrogen-bonded polycatenation non-covalent organic framework (HOF-FJU-52), exhibiting diverse and reversible RS behaviors with the high performance. Triggered by the external stimulus of electrical field $E$ at room temperature, HOF-FJU-52 has excellent resistive random-access memory (RRAM) behaviors, comparable to the state-of-the-art materials. When cooling down below 200 K, it was transferred to write-once-read-many-times memory (WORM) behaviors. The two memory behaviors exhibit reversibility on a single crystal device through the temperature changes. The RS mechanism of this non-covalent organic framework has been deciphered at the atomic level by the detailed single-crystal X-ray diffraction analyses, demonstrating that the structural dual-flexibility both in the asymmetric hydrogen bonded dimers within the 0D loops and in the infinite $\pi-\pi$ stacking column between the loops and chains contribute to reversible structure transformations between multistates and thus to its dual RS behaviors.

Non-covalent organic frameworks (NCOF), such as hydrogen-bonded organic frameworks (HOFs)[1-7], crystalline porous materials self-assembled from discrete organic building blocks through hydrogen-bonding interactions and other weak interactions such as π•••π interactions of about 0.7–40 kcal/mol, have attracted extensive attentions for their structural diversity, flexibility and reversibility, and their broad applications on gas separations[8,9], gas storage[10,11], photonics[12-14], proton conductivity[15-17], catalysis[18,19], chemical sensors[20,21] and enzyme encapsulation[22,23]. They have very good biomolecule compatibility and

solution processibility, and can be straightforwardly and easily recovered and regenerated simply through recrystallization[24]. Their structural flexibility and reversibility nature has enabled HOFs to exhibit multistate structures upon different external stimulus, providing great promise for their diverse functions and applications. For example, our recently reported HOF-FJU-1 with hydrogen-bonded dimers and infinite π–π stacking, self-assembled from a tetracyano organic molecule, shows a unique flexible-robust framework nature for thermo-regulatory gating effect in two very important $C_2H_4/C_2H_6$

[1]Fujian Provincial Key Laboratory of Polymer Materials, College of Materials Science and Engineering, Fujian Normal University, Fuzhou 350007 Fujian, China. [2]College of Chemistry, Fuzhou University, Fuzhou 350108, China. [3]These authors contributed equally: Shimin Chen, Yan Ju. ✉e-mail: zzhang@fjnu.edu.cn

and $C_3H_6/C_3H_8$ gas separations through the stretching vibration of flexible hydrogen bonds[25–27]. A flexible HOF-FJU-2 from a donor-π-acceptor molecule tetrabenzaldehyde carbazole shows adsorption-type stimulus-response for acetone vapor through reversible transformation between close and open phases, accompanied by reversible infinite π–π stacking switching, and eventually output turn-on fluorescence and color change signal visible to the naked eye[28]. It has been well established that the infinite π–π stacking arrays can provide effective carrier transport paths, assembly of such arrays into HOFs might provide a unique approach to developing multifunctional electronic and optical materials[29–31]. Furthermore, because such HOF structures will be very sensitive to external stimuli such as temperature and electrical field, it is expected that other functions such as resistive switching (RS) can also be realized other than those developed ones for the adsorption and photonics; however not well explored yet.

In order to design and develop resistive switching (RS) HOF materials, it is very important to understand the crystal structures of the different states of HOFs under the external stimulus, and thus correlate to their electrical bistability. Although some RS materials have been well developed, fundamental understanding of their RS behaviors at the molecular/atomic level has been very difficult because of their amorphous film nature[32,33]. The high crystalline nature of HOF materials has provided us the powerful platform to address this challenge, and thus to facilitate the discoveries of RS materials. In this work, we report a hydrogen-bonded polycatenation type framework HOF-FJU-52, $[(H_4L^2)(H_2L^1)][H_2L^1]\cdot(H_2O)_{12}$, self-assembled from the semirigid 1,1′-bis(4-carboxy-benzyl)−4,4′-bipyridinium dichloride $(H_2L^1Cl_2)$ and rigid pyrene-1,3,6,8-tetrayltetrakis(phosphonic acid)

$(H_8L^2)$ (Fig. 1a). HOF-FJU-52 is the 0D+1D hydrogen-bonded polycatenated sheet consisting of two units, a molecular loop $[(H_4L^2)(H_2L^1)]^{2-}$ and a ladder chain $[(H_2L^1)_n]^{2n+}$. In response to the stimulus of voltage and temperature, HOF-FJU-52 exhibits reversible transformations between multistate structures (Fig. 1 and Supplementary Table 4), which are subtle and unambiguous, enabling reversibly switchable RRAM and WORM memory behaviors on one single crystal device with high RS performance comparable to the state-of-the-art RS materials[32–34]. Detailed studies on RS mechanism by the single-crystal structure characterization clearly demonstrate the contribution of structural flexibility in asymmetric hydrogen bonds within the loop and infinite π–π stacking between the loops and the chains.

## Results

### Structure of HOF-FJU-52

The solvothermal reaction of $H_8L^2$ and $[H_2L^1]Cl_2$ in mixed MeOH/$H_2O$ (3:2, v/v) solution at 85 °C for 24 h gave orange bulk crystals of HOF-FJU-52, and its formula was substantiated by TGA, single- crystal x-ray diffraction (SCXRD) and elemental analysis (Supplementary Fig. 2 and Supplementary Table 4). HOF-FJU-52 crystallizes in the space group *Ccce*. It contains a 0D+1D polycatenated sheet (Fig. 2c), consisting of two units, that is, a ladder chain $[(H_2L^1)_n]^{2n+}$ (Fig. 2a) and a molecular loop $[(H_4L^2)(H_2L^1)]^{2-}$ (Fig. 2b). In the chain, there are half of crystallographically independent $H_2L^1$ molecule with *trans*-conformation, in which its 4,4′-bipyridine group acts as the "rung" and its 4-carboxyphenyl group acts as the "rail". $H_2L^1$ molecules link to each other through two O···H-C hydrogen bonds with O2···C29 distance of 3.067 Å to give a ladder chain along the *a* axis. In the loop, there are

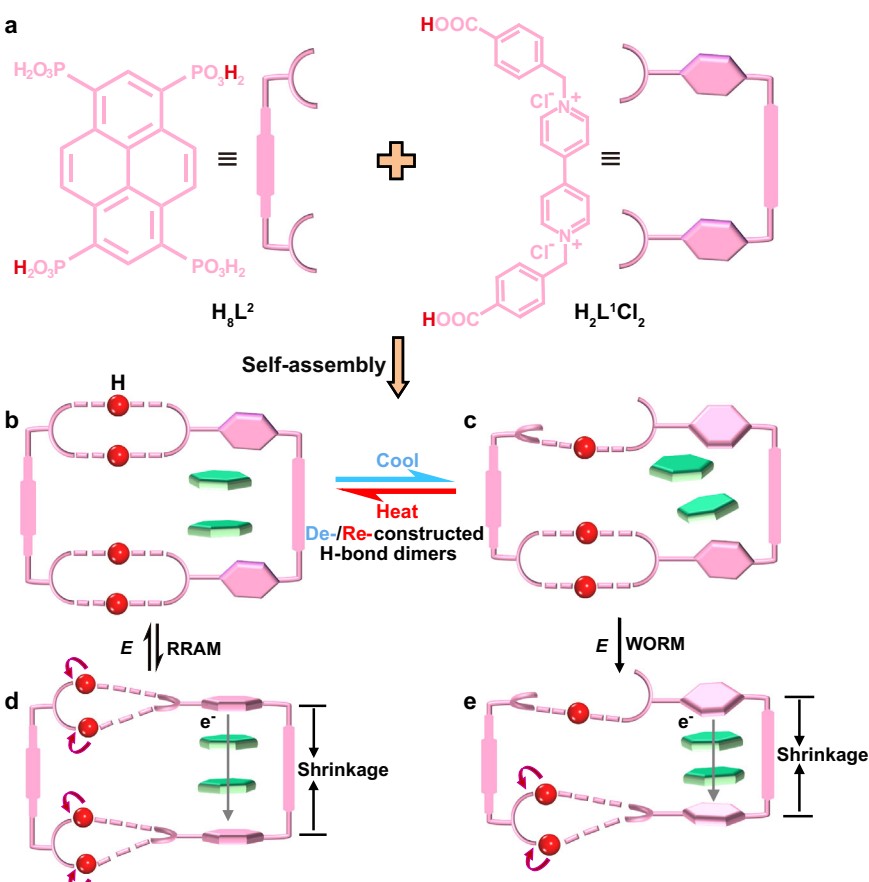

**Fig. 1 | Multistate structures in a hydrogen-bonded polycatenation. a** Two organic building units of the loop in HOF-FJU-52. **b, c** HOF-FJU-52 exhibiting the de-/re-construction of the asymmetric hydrogen bond dimers upon a stimulus of temperature. **d, e** Triggered by external electrical field *E*, structure transformations

in the loop with the resonant migration of shared protons of the H-bond dimers, and in the shrinkage of the π–π stacking column, undergoing reversible RRAM and WORM memory behaviors.

half of crystallographically independent $H_2L^1$ molecule with *cis*-conformation and half of $H_4L^2$ molecule. $H_2L^1$ acting as a "staple" employs its "pins" 4-carboxyphenyl groups to fasten $H_4L^2$ through two asymmetric O-H⋯O dimers between −COOH and −PO₃H groups with O9⋯O6 distance of 2.505 Å and O10⋯O8 distance of 2.627 Å to form a 0D molecular loop of $[(H_4L^2)(H_2L)]^{2-}$. Each 0D loop locks two adjacent ladders to generate 0D+1D hydrogen-bonded polycatenated sheet in the *ab* plane (Fig. 2c and Supplementary Fig. 3c). The sheets stack along the *c* axis through multiple π–π interactions between the pyridine of $H_2L^1$ and the pyrene of $H_4L^2$ from the two adjacent molecular loops, giving a three-dimensional HOF framework (Supplementary Figs. 3 and 4). The polycatenated assemblies in the previously reported HOFs are usually formed by interlocked 2D networks[35–37]. HOF-FJU-52 is the 0D+1D hydrogen-bonded polycatenated structures.

After the polycatenation, HOF-FJU-52 remains the solvent-accessible volume per unit cell 2894 Å³ (20% of the total unit cell volume), as calculated by the PLATON software[38], which is occupied by a large amount of lattice water molecules. After vacuum outgassing of the as-synthesized sample at 353 K for 12 h to remove the lattice water molecules, the activated HOF-FJU-52a sample can be obtained. The water vapor adsorption of the HOF-FJU-52a sample on 3Flex surface characterization analyzer monitors that at 294 K and $P/P_O = 0.95$, it can take water molecules with 11.3 per $H_4L^2$ (Supplementary Fig. 5), close to the value of its molecular formula. Also, its framework is well preserved after water vapor adsorption, as confirmed by the observed powder X-ray diffraction (PXRD) patterns (Supplementary Fig. 6).

Notably, the two couples of staggered 4-carboxyphenyl groups from *cis*- and *trans*-$H_2L^1$, that is "pin" and "rail", respectively, form a π–π stacking column along the *b* axis (Fig. 2d). The π•••π interactions with the centroid-centroid distance d1 of 3.451 Å are observed between the "pins" and the "rails" from the loop and the ladder, while (d2) of 3.464 Å are formed between two "rails" from the adjacent ladders (Supplementary Fig. 3a). And the dihedral angle between the phenyl ring and −COOH plane ($\theta1$) in the "pin" group is 4.28°, which is obviously larger than the dihedral angle of 1.78° in the "rail" group. This is mainly because the planarity of "pin" groups is simultaneously controlled by the hydrogen-bonding interactions in the asymmetric hydrogen bonds dimers and the π•••π interactions in the staggered "pin" and "rail" groups. Such a frustration may destroy the co-planarity of phenyl ring and −COOH in "pins" and hamper the π–π stacking, resulting in the larger inter-centroid distance between the two neighboring "pins" (d3) of 3.540 Å compared to the values of d1 and d2. The length of the packed π–π stacking column (d4) is 23.966 Å.

## SCSC transformation upon altering temperatures

Occasionally, we found an obvious temperature effect on the HOF structures. First, we explored changes in the single crystal structures upon altering the temperatures from 293 to 100 K (Fig. 3a). The temperature dropped from room temperature to 100 K at intervals of 50 K and then rose back to room temperature at the same interval. Eight crystal structures, termed HOF-FJU-52-XK(-R), can be obtained, in which X (X = 293, 250, 200, 150 and 100) and R represents the temperature and the reverse heating, respectively (Supplementary Fig. 7 and Supplementary Table 4). All of the crystals have a similar structure to HOF-FJU-52, featuring 0D+1D hydrogen-bonded polycatenation. At 200 K, an obvious SCSC transformation occurs with

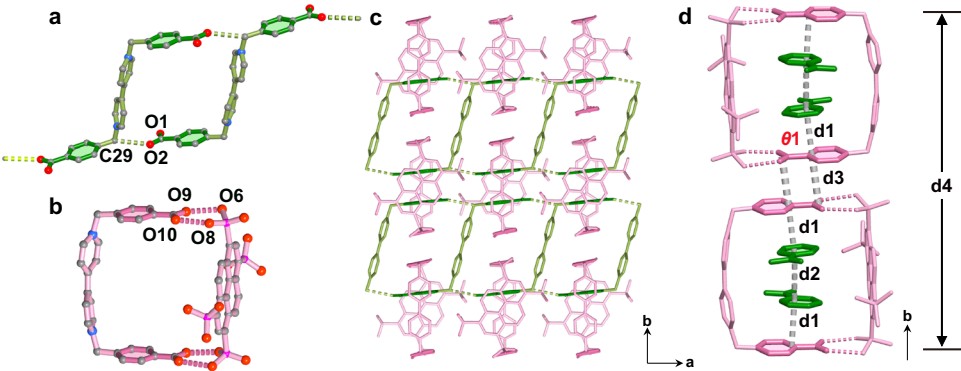

**Fig. 2 | Crystal structure of HOF-FJU-52. a** Ladder chain $[(H_2L^1)_n]^{2n+}$; **b** molecular loop $[(H_4L^2)(H_2L^1)]^{2-}$; **c** the 0D+1D→2D polycatenated sheet; **d** the infinite π–π stacking from staggered benzoic groups. Color code: P = magenta, C = dark gray, O = red, N = blue. H atoms and guest water molecules have been omitted for clarity.

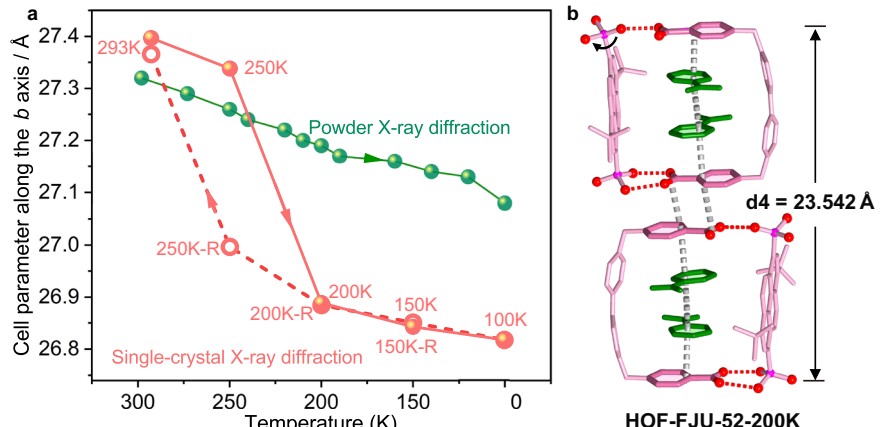

**Fig. 3 | Temperature effect on the HOF structures. a** An obvious hysteresis loop is observed in the function of the cell parameters *b* with the temperatures from single crystal XRD data, but no similar phenomenon from the VTPXRD data. Solid and open symbols represent cooling and heating process, respectively. **b** The variation of the asymmetric hydrogen bond dimers and π–π stacking column at 200 K.

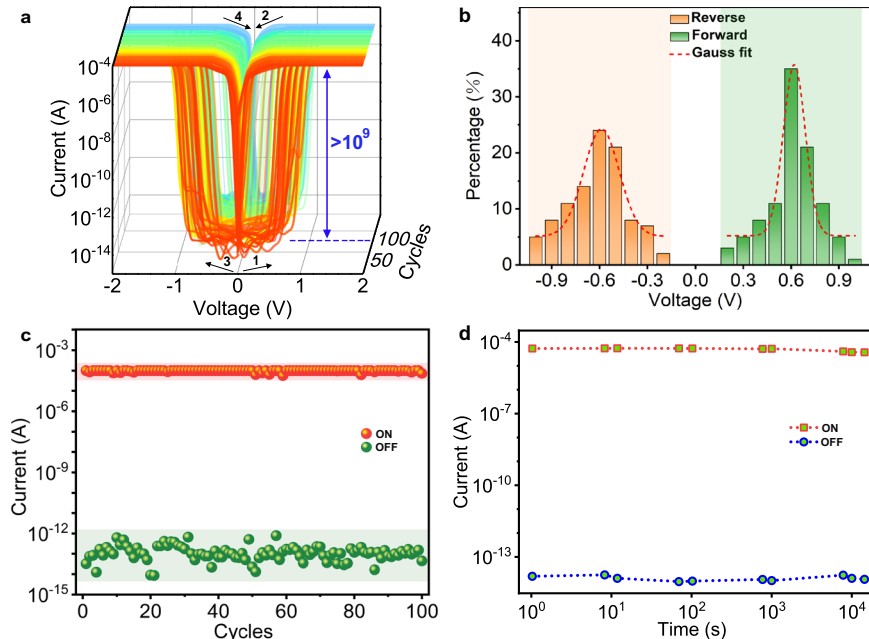

**Fig. 4 | Electrical performance of HOF-FJU-52 single crystal device along the *b* axis at room temperature. a** Semilogarithmic plot of current–voltage (*I–V*) characteristics of the device over 100 consecutive cycles. The arrows indicate sweeping direction while the numbers represent sweeping sequence. $I_{CC}$ stands for the compliance current ($1.0 \times 10^{-4}$ A). **b** Gaussian fitting curves of the distribution of the set voltage for forward and reverse voltage sweeps. **c** Endurance performance. **d** Retention performance of the two resistance states over 14,400 s. The sample resistance was read at 0.1 V.

the space group from *Ccce* in HOF-FJU-52 and HOF-FJU-52-250K to *C2cb* in HOF-FJU-52-200K. Especially, both the asymmetric hydrogen bond dimers and π–π stacking column display remarkable variations at 200 K (Supplementary Fig. 8). Above 200 K, the 0D loops in the HOFs can keep the hydrogen bond dimers in both sites. Upon altering the temperature from 200 to 100 K, the HOFs exhibit the destruction of half the hydrogen bond dimers in the 0D loops, accompanied by the rotation of one phosphonic acid group (Fig. 3 and Supplementary Fig. 9). Meanwhile, the π–π stacking column exhibits continuous shrinkage as the temperature decreases, consistent with the reduced cell unit parameters *b*. When it drops to 200 K, HOF-FJU-52-200K shows a sharp shrinkage with the rate of 0.009 Å per Kelvin, about nine times of the values compared to the values at the other temperatures. Oppositely, during the reverse heating process, no obvious SCSC transformation occurs at 250 K. The HOF with the original hydrogen-bonded dimer can be reconstructed after being placed at room temperature for 5 h, and the space group *Ccce* can also be restored. Meanwhile, the π–π stacking column can be expanded to the original state at room temperature, corresponding to the recovered cell unit parameters *b* (Supplementary Fig. 7a). Obviously, the recovery during the heating process only occurs at room temperature, lagging behind 250 K. As a result, HOF-FJU-52 is the single crystal showing a reversible hysteresis loop in the function of the cell parameter *b* with temperatures.

In order to clearly explain the hysteresis loop in the single crystal structures, we used the experimentally measured structures to calculate the energy difference between HOF-FJU-52 and HOF-FJU-52-200K using first-principles DFT calculations. As expected, the total energy of HOF-FJU-52-200K is −4019.69 eV, which is about 6.97 eV lower than that of HOF-FJU-52, implying that the HOF-FJU-52-200K is energetically more stable than the HOF-FJU-52. In another word, the structural transformation with the recovery of space group from *C2cb* to *Ccce* is likely to occur difficultly from a thermodynamic viewpoint; thus, no obvious SCSC transformation occurs even at 250 K during the reverse heating process and results in the hysteresis loop in the single crystal structures upon altering the temperatures.

Next, we obtained the in situ variable-temperature powder XRD (VTPXRD) from room temperature to 100 K, which almost remains unchanged. They only show a very slow shift of the (0 8 0) peaks to high angles (Supplementary Fig. 10), meaning a subtle shrinkage of the π–π stacking columns. The shrinkage rate is 0.0011 Å per Kelvin, as calculated from the slope in Supplementary Fig. 7b. No similar reversible hysteresis loop was observed in its VTPXRD experiments (Fig. 3a), which may perhaps be due to the quench effect as the fast cooling rate of 2 K/min during the test.

## RS behaviors at room temperature

The RS behaviors on the single crystal of HOF-FJU-52 device were measured by a typical current–voltage (*I–V*) curve in the ambient atmosphere. A compliance current (CC) preset of $1.0 \times 10^{-4}$ A was kept limited to prevent a complete breakdown of the device during the setting process. As evidenced in Supplementary Fig. 11, no RS effect is observed for HOF-FJU-52 along the *a* or *c* axis even when the voltage is applied up to 20 V. However, when the voltage is swept in the direction of 0 V → 2 V → 0 V → −2 V → 0 V along the *b* axis, HOF-FJU-52 exhibits interesting nonvolatile and electroforming-free electrical bistability (Fig. 4). It shows ultralow leakage currents (-0.1 pA) at the high resistance state. The set voltage is mainly distributed around ±0.6 V, lower than the values for most of RS materials. The ON/OFF ratio of HOF-FJU-52 can reach up to $10^9$, to the best of our knowledge, which is higher than the values for the most excellent crystalline materials[34] and comparable to the values for the most excellent organic[32] or inorganic materials[33] (Supplementary Fig. 12 and Supplementary Tables 1–3). The RS behavior of HOF-FJU-52 has good uniformity for more than 100 consecutive cycles (Fig. 4a–c). Both high resistance state (HRS) and low resistance state (LRS) can be retained for at least 14,400 s after removing the power supply (Fig. 4d), indicating a good data storage ability for the nonvolatile memory device. Our device also exhibits RS behaviors if exposed to various solvents and humidities, even under conditions of annealing at 500 K (Supplementary Fig. 13). These results suggest that the HOF-FJU-52 has great potential in the nonvolatile memory.

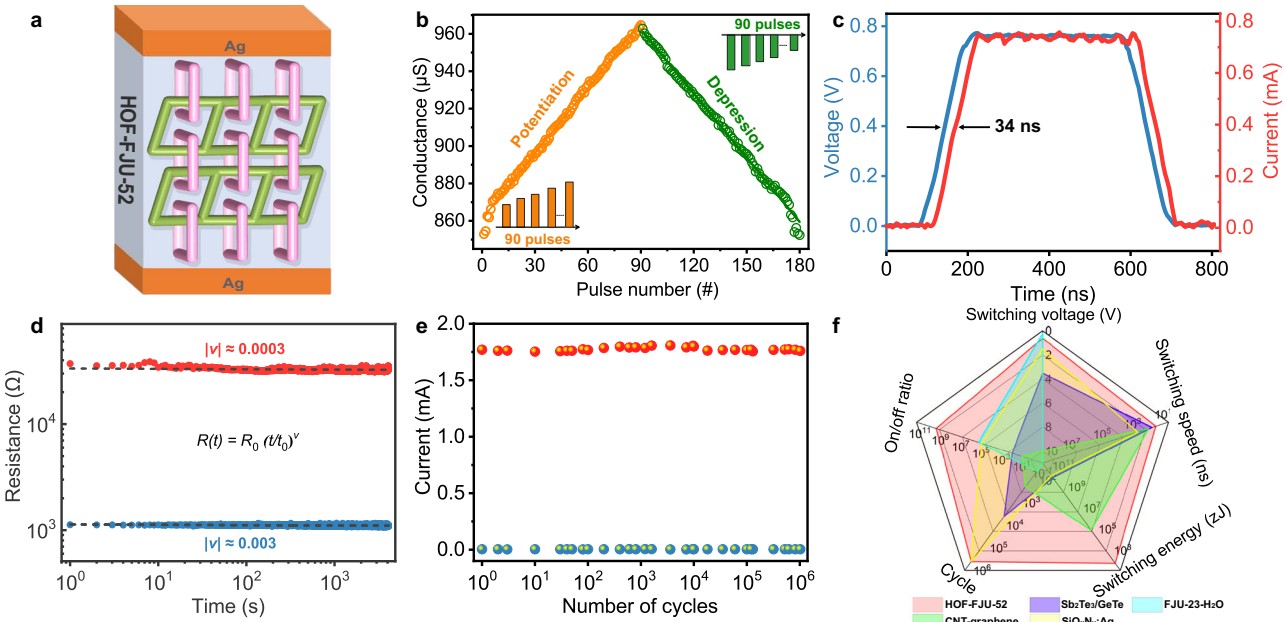

**Fig. 5 | Analog switching characteristics of the HOF-FJU-52 single crystal device along the *b* axis at room temperature. a** Schematic illustration of Ag/HOF-FJU-52/Ag device. **b** Potentiation and depression process measured by stepwise voltage pulses. **c** Current response with a fast voltage pulse. **d** Ultralow resistance drift for the HRS and LRS state of the HOF-FJU-52. $R_O$ is the resistance at an arbitrary time $t_O$, and $t$ is the time after the last switching event. **e** Pulse endurance test of the HOF-FJU-52 consistently. The programming was carried out with 60 ns voltage pulses of −0.1 and 1.9 V to RESET and SET, respectively. **f** Radar chart of comprehensive operation performance evaluation among some representative RS materials, as presented in Supplementary Tables 1–3.

Furthermore, we investigated the analog switching characteristics of biological synapses for HOF-FJU-52, whose potentiation and depression are regarded as two fundamental functions for neuronal communication[39], as schematized in Fig. 5a. For potentiation, the pulse amplitude increases from 0.15 to 1.87 V with 20 mV steps; for depression, the pulse amplitude decreases from 1.85 to 0.11 V with 20 mV steps (pulse width/interval: 200 ns/2 μs). Thus, a consecutive stepwise voltage pulse sequence consisting of 90 positive and 90 negative pulses is applied for the measurement (Fig. 5b). Its conductance modulation is symmetric and linear, suggesting that our device has great potential for future neuromorphic computing. When a voltage stimulus with the amplitude of 0.8 V and pulse width of 500 ns has been applied onto our device, an immediate response in the device current increasing to 0.74 mA can be observed with a write speed as far as 34 ns (Fig. 5c). The value of the switching energy per set transition ($E_{SET}$) can be calculated as $E_{SET} = I_{LRS} \times U_{SET} \times t_{SET}$. Although the higher electrical noise of the semiconductor parameter analyzer does not allow measurement of $t_{SET}$ when using $I_{CC}$ = 110 pA in Supplementary Fig. 14, $E_{SET}$ can be estimated for $I_{CC}$ < 1 nA by using the switching time obtained in Fig. 5c. This assumption is reasonable[40] because the switching time $t_{SET}$ from $I_{HRS}$ to $I_{CC}$ will surely be shorter than the value of 34 ns between $I_{HRS}$ and 0.74 mA. Consequently, the estimated switching energy even comes down to ~374 zJ, much lower than biological synapse (1–100 fJ per synaptic event) (Supplementary Fig. 14)[41]. By fitting the function of the HRS and LRS resistances with time, which can be obtained after reset/set pulses with the amplitude of −0.1/1.9 V and pulse width of 60 ns were applied to our device, we can monitor the resistance drift behaviors for its HRS and LRS with drift coefficients of 0.0003 and 0.003, respectively (Fig. 5d). The two coefficients are very tiny, about an order of magnitude lower than the values of conventional phase change random-access memory[42]. As shown in Fig. 5e, pulse-mode test suggests that the HOF-FJU-52 device can be switched reversibly and stably with set and reset pulses for more than $10^6$ cycles, which is indeed competitive with that of the commercial flash memories[43]. Fast switching speed, low switching energy, long cycling endurance, high ON/OFF ratio and low switching voltage are important

performance indicators for the RS materials. As shown in Fig. 5f and Supplementary Tables 1–3, the indicators above are comprehensively excellent for HOF-FJU-52. Although several RS materials including phase change chalcogenide $Sb_2Te_3$/GeTe thin-film[44], inorganic oxide $SiO_xN_y$:Ag thin-film[45], organic CNT-graphene composited thin-film[46], and inorganic-organic hybrid FJU-23-$H_2O$ single crystal[47] show one or more good performance indicators, these metrics are rarely achieved simultaneously in a single material.

To further confirm that the RS behaviors indeed correspond to the carrier transport on the framework rather than the ionic migration between the guest water molecules, we tested the RS performance of the activated HOF-FJU-52a device under vacuum at 353 K. It remains the RS behaviors with high ON/OFF ratio (~$10^9$), ultrafast switching speed (~33 ns), ultralow switching energy (363 zJ), and excellent endurance (~$10^6$) (Supplementary Figs. 15 and 16), indicating that RS behaviors are independent of the guest water molecules. Thus, desolvated HOF-FJU-52a represents the framework material for which microporosity and RS behaviors coexist.

## RS behavior upon altering temperatures

The reversible structural transformations in SCSC fashion inspired us to further investigate the temperature effect on its memory behaviors of one same single crystal device, as shown in Fig. 6 and Supplementary Figs. 17 and 18. Upon cooling, HOF-FJU-52-250K keeps the similar RRAM behavior to HOF-FJU-52, accompanying with the increasing set voltage from 0.6 to 1 V (Supplementary Fig. 17). Interestingly, when cooling down to SCSC transformation temperature of 200 K, HOF-FJU-52-200K shows a write-once-read-many-times memory (WORM) behavior instead of RRAM (Fig. 6a). The device switches from HRS to LRS with an ON/OFF ratio up to $10^9$ at a switching threshold voltage of about 2.42 V (sweep 1) during the *I–V* measurement. It cannot be switched back to its original HRS by a subsequent scan at a repeat positive voltage (sweep 2). It can maintain LRS even after applying a reverse voltage sweep (sweep 3 and sweep 4), suggesting the inerasable data storage behavior. Upon the recovered positive voltage, the LRS can also remain in sweep 5 and sweep 6, indicating a generation of

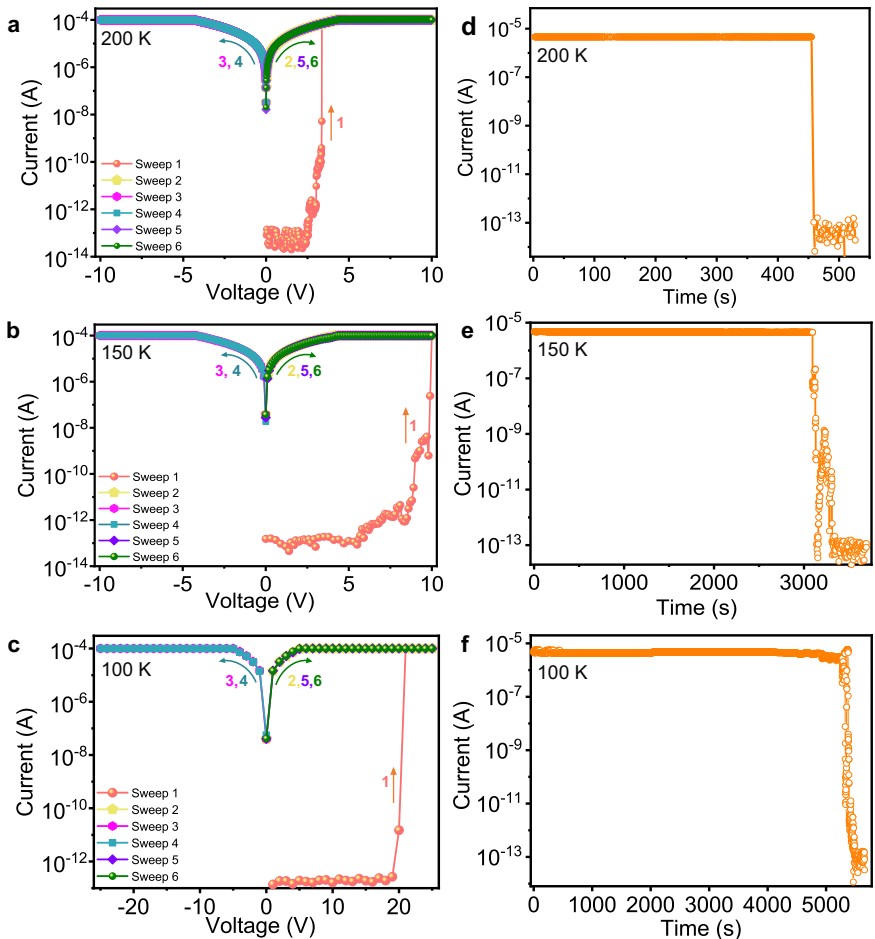

**Fig. 6 | WORM performance of our HOF device upon cooling down below 200 K.** WORM performance (**a**–**c**) and retention time read at 0.3 V (**d**–**f**) of our HOF device at 200, 150 and 100 K.

HOF-based WORM device. No significant degradation in the current for the high-conductivity state can be sustained for 458 s during the readout test (Fig. 6d). As the temperature drops to 150 K and 100 K, the device still keeps the WORM behavior, but giving an increase in set voltage (Fig. 6b, c) and an extension of retention time (Fig. 6e, f). Upon reverse heating, the device still shows the WORM behavior even at 250 K (Supplementary Fig. 18). In addition, RRAM behavior can be recovered if we place the device at room temperature for 5 h (Supplementary Fig. 19). Thus, upon cooling and heating, the memory behaviors of our single crystal HOF device can be reversibly switched between the RRAM and WORM, corresponding to the structural SCSC transformation with hysteresis loop, which has never been reported before.

## RS mechanism

In order to deeply reveal the RS mechanism, the multistate structures under the stimulus of various dc were explored on the same HOF-FJU-52 single crystal by SCXRD, attenuated total reflectance infrared (ATR-IR) and X-ray photoelectron spectroscopy (XPS). Firstly, we studied the mechanism of RRAM behavior at room temperature. In this condition, HOF-FJU-52 is a nonvolatile memory with a retention time of 14,400 s, long enough to complete the structural data collection by SCXRD. The structures of HOF-FJU-52-0.6V and HOF-FJU-52-(-)0.1V were obtained under the stimulus of 0.6 V and succedent −0.1 V voltages (Fig. 7 and Supplementary Fig. 21), respectively. In comparison with the pristine one, no substantial change has been observed in the two structures, except for some subtle changes in the asymmetric hydrogen bond dimers and the infinite π–π stacking from the staggered

4-carboxyphenyl groups. Viewed from the difference Fourier maps for the asymmetric hydrogen bond dimers in the pristine structure (Fig. 7a), the two protons are located nearly at the center position between two O atoms from the phosphonic and "pin" carboxylic acids within the loops, meaning protons are equally shared by O atoms in the dimers. Interestingly, the two shared protons migrate toward the positions near the phosphonic acid group but far away from the "pin" O atoms in HOF-FJU-52-0.6V (Fig. 7b), while the two protons migrate back to the center position of the hydrogen bond dimers in HOF-FJU-52-(-)0.1V (Fig. 7c), indicating that the voltage can induce the resonant migration of proton within the dimers. Supplementary Fig. 20 shows the ATR-IR difference spectra obtained before and after applying the voltage stimulus. Several broad and weak bands observed in the 2850–2350 cm$^{-1}$ range can be attributed to the ν(PO-H) and 2δ(POH) vibrations[48]. A very broad band centered at 3355 cm$^{-1}$ can be assigned to the stretching vibrations of H-bonded water molecules and CO-H stretching vibrations. Considering that the chemical environment of aromatic CH groups is not significantly affected during the RS state conversion process, the ν(C-H) bands at 3050 and 3121 cm$^{-1}$ can be utilized as the internal standard peaks. After the application of the set voltage, the broad and weak ν(PO-H) vibration bands of LRS (blue) are enhanced; meanwhile, its ν(CO-H) bands are slightly decreased. Upon applying the reset voltage, the intensities of ν(PO-H) bands and ν(CO-H) bands go back to their original levels, further indicating that the RS mechanism is related to the resonant migration of proton within the hydrogen bond dimers. XPS spectra of O1s of phosphonic and carboxylic acid groups further suggest the proton migration triggered by voltage, which are fitted using Shirley background subtraction and

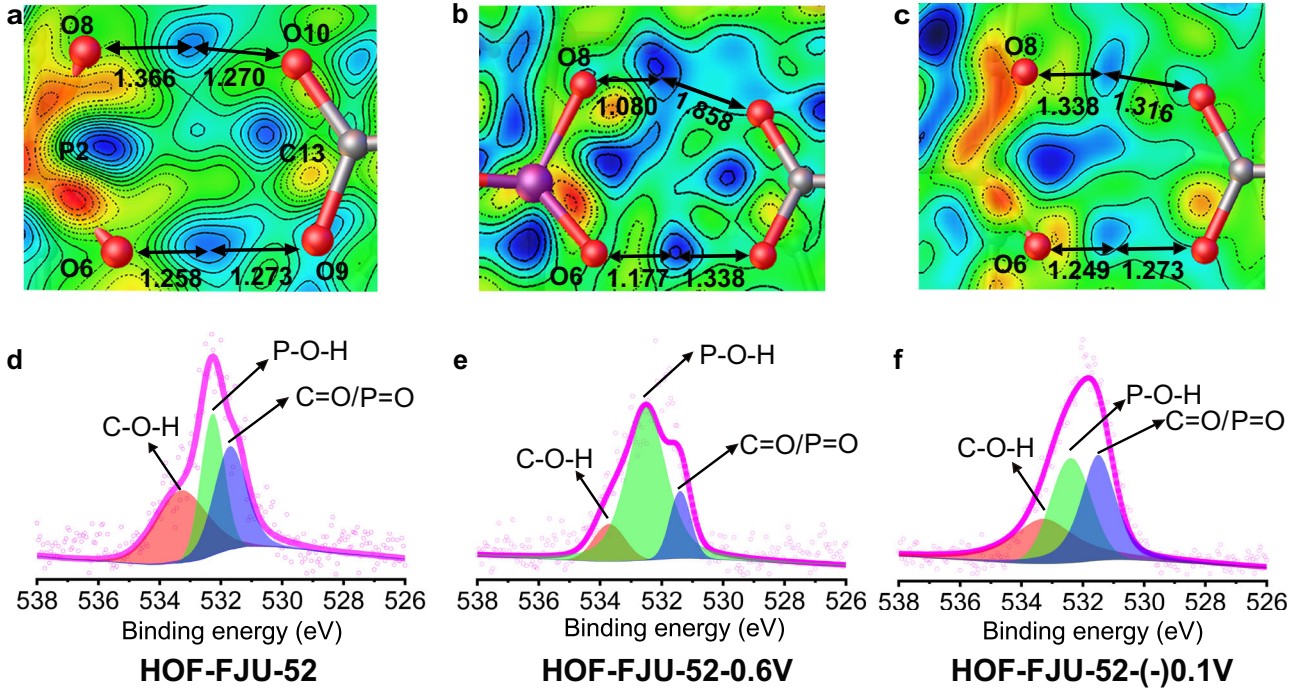

**Fig. 7 | The resonant migration of shared protons of the H-bond dimers under DC voltages. a–c** The migrations of the electron density peaks between phosphonic and carboxylic acid groups under DC voltages based on the difference Fourier maps. The unit for the distances between electron density peaks and oxygen atoms are given in Å. **d–f** X-ray photoelectron spectroscopy of O1s of phosphonic and carboxylic acid groups.

Gaussian functions. The O1s spectra in the pristine HOF are deconvoluted into three peaks at 531.44, 532.27 and 533.47 eV, corresponding to C=O/P=O, P-O-H, C-O-H, respectively[49,50], as confirmed by the XPS spectra of O1s of $H_2L^1Cl_2$ and $H_8L^2$ (Supplementary Fig. 22). Notably, an area ratio between the P-O-H and C-O-H components in the pristine HOF is near 1 (Fig. 7d). The ratio increases to 7 in HOF-FJU-52-0.6V (Fig. 7e), which suggests the increasing contents of P-O-H owing to the migration of shared protons from the center position to the P-O end. And the area ratio goes back to 1 in HOF-FJU-52-(-)0.1V after the reset voltage is applied (Fig. 7f), indicating that the protons migrate back to the center position of the hydrogen bond dimers. The XPS results agree very well with those from the SCXRD and ATR-IR spectra, further confirming the reversibly resonant migration of the shared protons in the asymmetric hydrogen bond dimers. Meanwhile, the distance d4 slightly shrinks from 23.966 to 23.929 Å in HOF-FJU-52-0.6V. And its dihedral angle $\theta$1 between benzene and carboxylic groups decreases from 4.28° to 2.91° (Supplementary Fig. 21a, b). This is probably because the deprotonation of the "pin" carboxylic groups triggers the dispelling of the "pin" frustration to decrease dihedral angle $\theta$1, leading to an increase of π-conjugation. The shorted π–π stacking favors the through-space electron transfer, resulting in the LRS behavior. As the protons migrate back to the original center position upon −0.1 V DC voltage applied, the dihedral angle $\theta$1 and d4 nearly come back to the initial state; thus, the HRS behavior is recovered in HOF-FJU-52-(-)0.1V. Therefore, structural reversibility in resonant migration of shared protons of the asymmetric H-bond dimers and in breathing effect of the π–π stacking column endows our HOF with reversible and nonvolatile RRAM behaviors at room temperature. Secondly, we also investigated the WORM mechanism in the crystal of HOF-FJU-52-100K-19V, in which the temperature was cooled down to 100 K and the voltage was raised up to 19 V. Although there is only one hydrogen-bonded dimers in 0D loop, the resonant migration of shared protons to phosphonic groups and the shrinkage (0.068 Å) of π–π stacking column are also observed, leading to the LRS behavior upon the external stimulus of electrical field E (Supplementary Fig. 23). The

destruction of half the hydrogen bond dimers may perhaps to a certain extent destroy the reversibility of the electrostructures of HOF-FJU-52 at the low temperature down from 200 K to 100 K, leading to the WORM behavior, instead of RRAM one at room temperature.

## Discussion

The crystalline porous materials exhibit potential as a platform for designing resistive switching materials. However, the relatively high bond energy of coordination bonds and covalent bonds in metal-organic frameworks[51–53] and covalent organic frameworks[54–56] making it difficult to form responsive backbones for effective resistive switching. The hydrogen-bonded interactions between host and guest molecules may provide a strategy for realizing resistive switching behavior, as observed in our previously reported MOF (FJU-23-H$_2$O)[47]. However, the difficulty of accurately controlling hydrogen-bonded interactions between host and guest molecules hinders the further development of research. We proposed the strategy of asymmetric hydrogen-bonding sites to obtain HOF-FJU-52, which is formed by hydrogen-bonding assembly between −COOH and −PO$_3$H groups. Being the 0D+1D hydrogen-bonded polycatenation, it features the 0D loops with asymmetric hydrogen bond dimers and infinite π–π stacking between the loops and the chains. Under temperature and electric field effects, asymmetric hydrogen bonds and π–π stacking change synergistically, resulting in 12 different structures and effectively regulating the resistive switching performance.

Besides, at room temperature, our HOF device exhibits comprehensively excellent RRAM performance indicators with ultrafast switching speed (~34 ns), ultralow switching energy (~374 zJ), excellent endurance (>$10^6$ cycles) and ultrahigh ON/OFF ratio (~$10^9$). These metrics are rarely achieved simultaneously in a single material, although several inorganic RS materials[44,45] show one or more good performance indicators. On the other hand, high-density storage is another great important performance pursued by molecular memory. Considering one infinite π–π stacking column in HOF-FJU-52 as memristor cell[47], a single crystal device with nanometric form accuracy

for future manufacturing can be realized with an ultrahigh storage density of 70.82 TB/cm$^2$, 400 times higher than that for traditional mechanical hard disk[43]. Unlike the integration of the traditional inorganic RS device by using lift-off process[57], the integration of single crystal nanotechnology still requires further exploration of wire bonding technology. More interestingly, HOF-FJU-52 exhibits diverse RS behaviors. Upon cooling and heating, the RS behaviors of HOF-FJU-52 device can be switched between the RRAM and WORM. There is the coexistence of microporosity and diverse RS behaviors in the HOF-FJU-52a, which may pave a way to construct guest stimuli-responsive memristor with integrated technologies for sensing, storage, and computing.

At last, it is worth mentioning that the single crystal RS device provides a powerful tool to describe a clear picture of the change between the multistate structures deciphered by atomic level characterizations. After performing SCXRD analysis, we discovered that the reversible RRAM and WORM behaviors upon altering temperatures are contributed from SCSC structural transformation, which is characterized by a reversible hysteresis loop of the unit cell parameters. We also observed that the RS behaviors originate from structural reversibility in the resonant migration of shared protons of the asymmetric H bonds and in the breath effect of the π−π stacking column. But no similar phenomenon is observed from the VTPXRD data. In order to meet the requirements of large-scale integrated circuits for industrial applications, thin-film media are mostly employed in the RS devices[32,33,58−67], while single crystal media are usually ignored. Various RS mechanisms of the amorphous film, such as ion migration, interfacial reaction, charge trapping/de-trapping and thermochemical reaction, significantly rely on active metal electrodes and metal nanoparticles[32]. Although such mechanisms have been investigated by means of theoretical calculations, XRD and other characterization technologies, such as conductive atomic force microscopy (CAFM) or transmission electron microscopy (TEM), which is still lack of atomic level characterizations to reveal the changes of intrinsic structures upon the external stimulus of the electrical field. Therefore, the single-crystal device provides a powerful tool for mechanism interpretation and also offers insights into the design of advanced resistive switching materials. In all, our work will promote the multifunctionalization of HOFs and provide great opportunities for resistive switching materials.

## Methods

### Materials
1,3,6,8-Tetrabromopyrene (95%, Adamas), 4,4′-Bipyridine (98%, Adamas), 1,3-Diisopropylbenzene (96%, Adamas), Triethyl Phosphite (98%, Adamas), Ethyl Acetate (99.5%, SCRC), Hexane (99.5%, SCRC), Methyl Alcohol (99.5%, SCRC), 4-Chlorobenzoic Acid (99%, Adamas).

### Synthesis of $H_2L^1Cl_2$
$H_2L^1Cl_2$ was synthesized according to the previously published procedures[68]. A mixture of 4,4′-bipyridine (6.247 g, 40 mmol) and 4-(chloromethyl)benzoic acid (13.647 g, 80 mmol) was dissolved in 50 mL of DMF and then stirred at 120 °C under nitrogen for 4 h. After the mixture was cooled to room temperature, and the resulting precipitate was separated by filtration, washed with DMF, and then dried under vacuum to obtain $H_2L^1Cl_2$ in the form of a white powder (yield: 54%). The observed NMR peaks in $D_2O$ matched the literature[69].

### Synthesis of $H_8L^2$
$H_8L^2$ was prepared using the modified workup described in the literature[70]. Under $N_2$ atmosphere, a mixture of 1,3,6,8-tetrabromopyrene (2.5892 g, 5 mmol), anhydrous nickel (II) chloride (0.26 g, 2.0 mmol), and diisopropylbenzene (20 mL) was stirred at 190 °C for 30 min. P(OEt)$_3$ (4.652 g, 28 mmol) was slowly added over a period of 4 h, and the mixture was stirred at 190 °C for another 8 h. After cooling to room temperature, the solvent was removed by rotary

evaporation and the crude material was recrystallized using ethyl acetate and n-hexane, and then dried in vacuo to give the octaethyl pyrene-1,3,6,8-tetrayltetrakis(phosphonate) (PE) as a yellow powder (yield 66%). A solution of PE powder in 6 mol/L HCl was refluxed for 10 h and then dried under vacuum to obtain $H_8L^2$ in the form of an aqua powder (yield 60%). The observed NMR peaks in $D_2O$ matched the literature[70].

### Synthesis of HOF-FJU-52
A mixture of $H_2L^1Cl_2$ (3.8 mg, 0.0075 mmol), $H_8L^2$ (1.31 mg, 0.0025 mmol), 5 mL CH$_3$OH/H$_2$O (1.5:3.5, v/v) and 2 mL *N,N*′-dimethylacetamide (DEF) solution was stirred for 5 min. Then, the solution was transferred to a 23-ml glass jar and heated to 85 °C. After 24 h, the system was cooled to room temperature, and orange bulk crystals were obtained. The yield of the HOF-FJU-52 was 25% based on $H_8L^2$. Cal: C, 51.45%; H, 4.95%; N, 3.52%; found C, 50.99%; H, 4.95%; N, 3.37%. The thicknesses of HOF-FJU-52 single crystals are in the ranges of 0.004−0.007 cm. Seeking a suitable approach to tune the thickness is still on the way. Some information on PXRD, TGA for HOF-FJU-52 is listed in Supplementary Figs. 2 and 6.

### Characterization methods
The thermogravimetric analysis (TGA) of the sample was conducted by a Mettler Toledo TGA/SDTA851e analyzer in $N_2$ with a heating rate of 5 K min$^{-1}$ from 30 to 600 °C. The elemental analyses (EA) of C/H/N were performed by using vario EL elemental analyzer. The powder X-ray diffraction (PXRD) was obtained by a PANalytical X'Pert powder diffractometer with Cu Kα radiation (λ = 1.54184 Å) at 40 kV and 40 mA over the 2θ range of 5° to 40°. *I*−*V* characteristics of single crystals were recorded on a Lakeshore probe station by using a precision Keithley 4200-SCS semiconductor parameter analyzer with DC module and pulse module. The XPS measurements were implemented on Thermo ESCALAB 250 spectrometer using nonmonochromatic Al Kα X-ray as the excitation source. All XPS spectra were recorded on one single crystal with an area of diameter 50 μm.

### Single-crystal X-ray structure determination
The single-crystal data were collected by the Agilent Technologies SuperNova Single-Crystal Diffractometer equipped with graphite monochromatic Cu Kα radiation (λ = 1.54184 Å). All the structures were solved by direct methods and refined on F$^2$ by full matrix least-square using Olex2 program[71]. All the non-hydrogen atoms were refined anisotropically. The hydrogen atoms between O6 and O9, O8 and O10 were added to the difference Fourier maps. Other H atoms on lattice water were added geometrically and refined using the riding model because of the lack of Q peaks in the appropriate position around O atoms.

The detailed crystallographic data and structure refinement parameters for HOF-FJU-52 is summarized in Supplementary Tables 4 and 5. Under various DC voltages, structure determinations were conducted by the following procedures. A pristine single crystal of HOF-FJU-52 was mounted onto the SuperNova Diffractometer for collecting diffraction data to get the structure of HOF-FJU-52 under ambient atmosphere. Next, this crystal was placed on the probe station with a Keithley 4200 system and DC voltage of 0.6 V was applied on HOF-FJU-52 along the *b* axis to obtain the structure of HOF-FJU-52-0.6V under ambient atmosphere. Then HOF-FJU-52-0.6V was also mounted onto the SuperNova Diffractometer for collecting diffraction data. Similarly, the DC voltage of −0.1 V was applied on HOF-FJU-52-0.6V along *b* axis to get the structure of HOF-FJU-52-(-)0.1V under ambient atmosphere. On the other hand, eight crystal structures, termed HOF-FJU-52-XK(-R) (X = 250, 200, 150, 100) can be obtained when the temperature drops from room temperature to 100 K at an interval of 50 K and then rises back to room temperature at the same interval. X represents the temperature tested, while R does the reverse heat.

Finally, the DC voltage of 19 V was applied on HOF-FJU-52-100K-HRS along the *b* axis to obtain the structure of HOF-FJU-52-100K-19V.

## Fabrication and characterization of Memory Devices

The glass substrates were precleaned sequentially with ethanol in an ultrasonic bath for 30 min. One of HOF-FJU-52 single crystal with approximate dimensions of 0.004 cm × 0.014 cm × 0.018 cm was selected and placed onto the glass substrate along the *b* axis. Silver conducting pastes were painted on the top and bottom sides of the crystal to obtain the single-crystal device. Our RS devices have been built with two Ag electrodes sandwiching our HOF materials (Ag/HOF-FJU-52 single crystal/Ag), the HOF material works as an active medium layer in resistive switching memory systems, as shown in Supplementary Fig. 24. Next, the glass was put into the Lakeshore CRX-VF Cryogenic Probe Station. Two tungsten tips were in direct contact with the silver conducting pastes (tips diameter <20 μm). A bias voltage was applied through the tungsten tip during the *I*–*V* measurements. An $I_{CC}$ preset of $10^{-4}$ A was kept limited to avoid a complete breakdown of the sample. The electrical performance tests at room temperature were conducted under dry ambient atmosphere. Other electrical performance tests under high/low temperatures were adjusted by a Lake-Shore 336-Temperature controller in a vacuum environment (-$10^{-4}$ torr) (Supplementary Fig. 24). The electrical properties on single crystal devices were measured by using a Keithley 4200-SCS semiconductor parameter analyzer with two modules: the sensitivity of DC mode of the Keithley 4200 system was 10 fA accomplished with 4200-SMU DC source unit. The pulse measurements were implemented using 4225-PMU waveform generator unit and 4225-RPM remote preamplifier/switch modules.

## Water vapor adsorption measurements

HOF materials (HOF-FJU-52) were degassed at 353 K under vacuum condition for 12 h at a Micromeritics 3Flex degas station before adsorption measurements. Water vapor adsorption isotherm of activated HOF-FJU-52a was measured on Micromeritics 3Flex surface characterization analyzer at 294 K.

## XPS test

X-ray photoelectron spectroscopy (XPS) was conducted on a Kratos Axis Ultra delay line detector (DLD) spectrometer equipped with a monochromated Al Kα X-ray source as the excitation source. All XPS spectra were recorded on one single crystal with an area of diameter 50 μm.

## ATR-IR experiments

The ATR-IR experiments were performed using a Nicolet iS50 ATR Infrared Spectrometer. We selected eight HOF-FJU-52 single crystals with appropriate dimensions and placed them above the ATR attachment to obtain the HOF-FJU-52 spectrum. Next, a second set of eight HOF-FJU-52-0.6V single crystals was prepared and placed above the ATR attachment to obtain the HOF-FJU-52-0.6V spectrum. Similarly, a third set of eight HOF-FJU-52-(-)0.1V single crystals was prepared and placed above the ATR attachment to obtain the HOF-FJU-52-(-)0.1V ATR-IR spectrum.

## Computational methods

The density functional theory (DFT) calculations were performed via the Vienna ab initio simulation package VASP (Version 5.0)[72,73], and the projector augmented wave (PAW) potentials were used to describe the electron-ion interactions. The generalized gradient approximation (GGA) of Perdew-Burke-Ernzerhof (PBE) exchange-correlation functional was employed. The kinetic cut-off energy of the plane-wave expansion was set to 400 eV, and only point was involved in the Brillouin zone integration. Moreover, the effects of van der Waals interactions were taken into account by adopting Grimme's DFT-D2 method. All the theoretical calculations were carried out by using the structure measured experimentally, which contains 1200 atoms in the unit cell with a composition of $C_{544}H_{432}N_{32}O_{160}P_{32}$.

## Data availability

All data supporting the findings of this study are available within this article and its Supplementary Information. These data can be obtained from https://doi.org/10.6084/m9.figshare.24591546. Cambridge Crystallographic Data Centre (CCDC) 2052476 for HOF-FJU-52, 2058177 for HOF-FJU-52-0.6V, 2058181for HOF-FJU-52-(-)0.1V, 2095208 for HOF-FJU-52-250K, 2095212 for HOF-FJU-52-200K, 2105287 for HOF-FJU-52-150K, 2105289 for HOF-FJU-52-100K, 2225349 for HOF-FJU-52-150K-R, 2225356 for HOF-FJU-52-200K-R, 2225743 for HOF-FJU-52-250K-R, 2225358 for HOF-FJU-52-293K-R, 2246710 for HOF-FJU-52-100K-HRS and 2246372 for HOF-FJU-52-100K−19V. These data can be obtained free of charge from https://www.ccdc.cam.ac.uk/structures/. Source Data are provided with this paper.

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

## Acknowledgements

This work was financially supported by the National Natural Science Foundation of China (nos. 22271046, 21975044, 21971038 and 22101050), and the Fujian Provincial Department of Science and Technology (nos. 2019L3004).

## Author contributions

S.C., Z.Z., S.X., and B.C. conceived the research idea and designed the experiments. S.C. and Y.J. performed the experiments and analyzed data. S.C., Y.J., Y.Y., F.X., Z.Y., H.Z., Y.L., Y.Z., S.X., B.C., and Z.Z. wrote the paper.

## Competing interests

The authors declare no competing interests.
