## [Peer review file · Nature Communications]

REVIEWER COMMENTS

Reviewer #1 (Remarks to the Author):

This is a very interesting work about hydrogen bound organic frameworks (HOFs) in the application of resistive switching (RS) behavior. HOF-FJU-52 is the first example of 0D+1D hydrogen-bonded polycatenation HOF. In response to the stimulus of electrical field E and temperature T , it exhibits reversible transformations between multistate structures, enabling reversibly switchable resistive random-access memory (RRAM) behavior and write-once-read-many-times memory (WORM) behavior on one single crystal device with high RS performance. Furthermore, the RS mechanism has been revealed at atomic level by single-crystal X-ray diffraction analyses. This work is expected to make a great impact on HOF chemistry and materials science, meanwhile to push the development of RS materials. I think it definitely deserves publication in Nature Communications after addressing the following questions:

1. To the best of my knowledge, the traditional inorganic RS materials have been extensively researched and applied in RS field. Thus I am curious about the advantages of developing emerging HOF materials for resistive switching performances compared to traditional inorganic RS materials?
2. I notice that the authors performed the RS measurements on single crystals. Is it possible to measure RS performances on pellets? Do you think the same results can be achieved if pellet samples are used?
3. It will help readers to understand the research background, if the authors can provide some information about the future prospects of resistive switching materials.
4. As shown in Figure 4, the authors claimed it was a RS device. But it seems not clear enough to me about the device structure. The author should clarify the role of HOF materials in the resistive switching devices in the experimental part.

Reviewer #2 (Remarks to the Author):

The manuscript by Z Zhang and coworkers describes the preparation of hydrogen-bonded polycatenation HOF-FJU-52 and investigation of its resistive switching properties. Employing the HOF single-crystal as the resistive switching material is novel and the resistive switching performance is nice. It offers the chance to monitor the state changes at different RS states. I would like to recommend acceptance for publication after addressing the following points.

- 1 The device on substrate is fabricated with a selected single-crystal. The single-crystal used is perfect for the device. However, these fabrication methods are not easy to obtain large amounts of devices. The authors should provide some potential approaches for future manufacturing, such as in situ methods should be tried.
- 2 How about the thickness of the active material? The thickness is tunable or not?
- 3 There is not strong evidence for the structure change during the RS state conversion. The authors have the opportunity to provide stronger characterization for the single-crystal based device. In this case, the mechanism will be much more clear.
- 4 How about the stability of the RS device upon the solvent, humidity, and temperature?
- 5 Typical reports on RS behaviors should be cited, COFs, porous polymers, etc.

Point-by-point Responses to Reviewers' Comments

Manuscript number: NCOMMS-23-27012

Title: Multistate structures in a hydrogen-bonded polycatenation HOF with diverse resistive switching behaviors

Response to Referees:

Reviewer #1 (Remarks to the Author):

This is a very interesting work about hydrogen bound organic frameworks (HOFs) in the application of resistive switching (RS) behavior. HOF-FJU-52 is the first example of 0D+1D hydrogen-bonded polycatenation HOF. In response to the stimulus of electrical field E and temperature T , it exhibits reversible transformations between multistate structures, enabling reversibly switchable resistive random-access memory (RRAM) behavior and write-once-read-many-times memory (WORM) behavior on one single crystal device with high RS performance. Furthermore, the RS mechanism has been revealed at atomic level by single-crystal X-ray diffraction analyses. This work is expected to make a great impact on HOF chemistry and materials science, meanwhile to push the development of RS materials. I think it definitely deserves publication in Nature Communications after addressing the following questions:

Response: Thank you very much for your positive comments.

Question 1: To the best of my knowledge, the traditional inorganic RS materials have been extensively researched and applied in RS field. Thus I am curious about the advantages of developing emerging HOF materials for resistive switching performances compared to traditional inorganic RS materials?

Answer: Yes, we have revised the corresponding contents in Main Text, Page 14.

Modification:

Main Text: Page 14, Paragraph 2.

Besides, at room temperature, our HOF device exhibits comprehensively outstanding RRAM performance indicators with ultrafast switching speed (~ 34 ns), ultralow switching energy (~ 374 zJ), excellent endurance ($>10^6$ cycles) and ultrahigh ON/OFF ratio ($\sim 10^9$). These metrics are rarely achieved simultaneously in a single material, although several inorganic RS materials^{44,45} show one or more good performance indicators.

Question 2: I notice that the authors performed the RS measurements on single crystals. Is it possible to measure RS performances on pellets? Do you think the same results can be achieved if pellet samples are used?

Answer: Yes, according to the valuable comments. Yes, we have conducted the measurements. But no obvious RS effect is observed for HOF-FJU-52 pellet sample, though we have tried several times. The RS effect of pellet sample may be greatly affected by the orientation of pellets. Our current work is focused on the performance of single crystals. The work about pellet will be further studied in the near future. We have appended relevant contents in Supporting Information, Page S27.

Modification:

Supporting Information: Page S27, Supplementary Fig. 25.

It is a pity that no obvious RS effect is observed for HOF-FJU-52 pellet sample, though we have tried several times.

Supplementary Figure 25 Electrical performance of the HOF-FJU-52 pellet device. The inset shows a microscopic image of the pellet sample configuration.

Question 3: It will help readers to understand the research background, if the authors can provide some information about the future prospects of resistive switching materials.

Answer: Yes, we have added the corresponding contents into Main Text in Page 14.

Modification:

Main Text: Page 14, Paragraph 2.

Considering one infinite π - π stacking column in HOF-FJU-52 as memristor cell⁴⁷, a single crystal device with nanometric form accuracy for future manufacturing can be realized with

an ultrahigh storage density of 70.82 TB/cm², 400 times higher than that for traditional mechanical hard disk⁴³. Unlike the integration of the traditional inorganic RS device by using lift-off process⁵⁷, the integration of single crystal nanotechnology still requires further exploration of wire bonding technology.

Question 4: As shown in Figure 4, the authors claimed it was a RS device. But it seems not clear enough to me about the device structure. The author should clarify the role of HOF materials in the resistive switching devices in the experimental part.

Answer: Yes, we have appended the experiment in Methods section in Main Text, Page 16 and Supporting Information, Page S26.

Modification:

Main Text: Page 16, Paragraph 3.

Silver conducting pastes were painted on the top and bottom sides of the crystal to obtain the single crystal device. Our RS devices have been built with two Ag electrodes sandwiching our HOF materials (Ag/HOF-FJU-52 single crystal/Ag), the HOF material works as an active medium layer in resistive switching memory systems, as shown in Supplementary Fig. 24.

Supporting Information: Page S26, Supplementary Fig. 24.

Supplementary Figure 24. The microscopic image of the experimental setup.

Reviewer #2 (Remarks to the Author):

The manuscript by Z Zhang and coworkers describes the preparation of hydrogen-bonded polycatenation HOF-FJU-52 and investigation of its resistive switching properties. Employing the HOF single-crystal as the resistive switching material is novel and the restive switching

performance is nice. It offer the chance to monitor the state changes at different RS states. I would like to recommend acceptance for publication after addressing the following points.

Response: Thank you very much for your very positive comments.

Question 1: The device on substrate is fabricated with a selected single-crystal. The single-crystal used is perfect for the device. However, these fabrication method is not easy to obtain large amount of devices. The authors should provide some potential approaches for future manufacturing, such as in situ methods should be tried.

Answer: Yes, we have appended the corresponding contents into Main Text in Page 14.

Modification:

Main Text: Page 14, Paragraph 2.

Considering one infinite π - π stacking column in HOF-FJU-52 as memristor cell⁴⁷, a single crystal device with nanometric form accuracy for future manufacturing can be realized with an ultrahigh storage density of 70.82 TB/cm², 400 times higher than that for traditional mechanical hard disk⁴³. Unlike the integration of the traditional inorganic RS device by using lift-off process⁵⁷, the integration of single crystal nanotechnology still requires further exploration of wire bonding technology.

Question 2: How about the thickness of the active material? The thickness is tunable or not?

Answer: Yes, we have added the thickness of the active material and revised the contents in Main Text, Page 15.

Modification:

Main Text: Page 15, Paragraph 3.

The thicknesses of HOF-FJU-52 single crystals are in the ranges of 0.004-0.007 cm. Seeking a suitable approach to tune the thickness is still on the way.

Question 3: There is not strong evidence for the structure change during the RS state conversion. The authors have the opportunity to provide stronger characterization for the single-crystal based device. In this case, the mechanism will be much more clear.

Answer: Thank you for this important question. The single-crystal based device can offer deep insights into mechanism interpretation during the RS and to guide the design of new RS materials. Within this context, 1) we have obtained 12 electrostructures by single-crystal X-ray diffraction analyses (SCXRD), revealing various RS behaviors for our HOF device,

under various electric fields and temperatures. 2) We have obtained the difference spectra of attenuated total reflectance infrared (ATR-IR) before and after applying the voltage stimulus, indicating that the RS mechanism is related to the resonant migration of proton within the hydrogen bond dimers. 3) We have observed X-ray photoelectron spectroscopy (XPS) of O1s of phosphonic and carboxylic acid groups for 3 electrostructures, confirming the reversibly resonant migration of the shared protons in the asymmetric hydrogen bond dimers. On the other hand, it is a pity that Raman and fluorescence spectroscopies cannot effectively distinguish the various electrostructures with very subtle changes (Figure R1 and R2), though we have attempted several times. We have appended the contents in Main Text, Page 12 and Supporting Information, Page 22.

Figure R1. Raman spectra of one same HOF-FJU-52 single crystal before and after the voltage stimulus at 0.6 V for 20 minutes.

Figure R2. Fluorescence spectra of one same HOF-FJU-52 single crystal before and after the voltage stimulus at 0.6 V for 20 minutes.

Modification:

Main Text: Page 12, Paragraph 1.

Supplementary Fig. 20 shows the ATR-IR difference spectra obtained before and after applying the voltage stimulus. Several broad and weak bands observed in the 2850-2350 cm^{-1} range can be attributed to the $\nu(\text{PO-H})$ and $2\delta(\text{POH})$ vibrations⁴⁸. A very broad band centered at 3355 cm^{-1} can be assigned to the stretching vibrations of H-bonded water molecules and CO-H stretching vibrations. Considering that the chemical environment of aromatic CH groups is not significantly affected during the RS state conversion process, the $\nu(\text{C-H})$ bands at 3050 and 3121 cm^{-1} can be utilized as the internal standard peaks. After the application of the set voltage, the broad and weak $\nu(\text{PO-H})$ vibration bands of LRS (blue) are enhanced, meanwhile its $\nu(\text{CO-H})$ bands are slightly decreased. Upon applying the reset voltage, the intensities of $\nu(\text{PO-H})$ bands and $\nu(\text{CO-H})$ bands go back to their original levels, further indicating that the RS mechanism is related to the resonant migration of proton within the hydrogen bond dimers.

Main Text: Page 17, Paragraph 3.

The ATR-IR experiments were performed using a Nicolet iS50 ATR Infrared Spectrometer. We selected eight HOF-FJU-52 single crystals with appropriate dimensions and placed them above the ATR attachment to obtain the HOF-FJU-52 spectrum. Next, a second set of eight HOF-FJU-52-0.6V single crystals was prepared and placed above the ATR attachment to obtain the HOF-FJU-52-0.6V spectrum. Similarly, a third set of eight HOF-FJU-52-(-)0.1V single crystals was prepared and placed above the ATR attachment to obtain the HOF-FJU-52-(-)0.1V ATR-IR spectrum.

Supporting Information: Page S22, Supplementary Fig. 20.

Supplementary Figure 20. ATR-IR spectra of eight HOF-FJU-52 single crystals before and after the

voltage stimulus at 0.6 V for 20 minutes.

Question 4: How about the stability of the RS device upon the solvent, humid, and temperature?

Answer: Yes, according to the valuable comments, we have tested the RS behaviors of the HOF-FJU-52 single crystal device upon the various solvents and humidities, even under conditions of annealing at 500 K, which has been appended in Main Text, Page 8 and Supporting Information, Page S15.

Modification:

Main Text: Page 8, Paragraph 1.

Our device also exhibits RS behaviors if exposed to the various solvents and humidities, even under conditions of annealing at 500 K (Supplementary Fig. 13)

Supporting Information: Page S15, Supplementary Fig. 13.

Supplementary Figure 13. RS behaviors of the HOF-FJU-52 single crystal device upon (a) annealed at

500 K in vacuum condition, (b) at 25 °C with different relative humidity, and (c) soaked in various solvents.

Question 5: Typical reports on RS behaviors should be cited, COFs, porous polymers, etc.

Answer: Yes, several typical reports on RS behaviors, COFs, porous polymers, etc., have been cited as refs. 60-67 in the revised Main Text, Page 20.

Modification:

Main Text: Page 20, refs. 60-67.

References

60. Sun, B. et al. Resistive switching memory performance of two-dimensional polyimide covalent organic framework films. *ACS Appl. Mater. Interfaces* **12**, 51837-51845 (2020).
61. Li, T. et al. 2D oriented covalent organic frameworks for alcohol-sensory synapses. *Mater. Horiz.* **8**, 2041-2049 (2021).
62. Li, C. et al. Towards high-performance resistive switching behavior through embedding a D-A system into 2D imine-linked covalent organic frameworks. *Angew. Chem. Int. Ed.* **60**, 27135-27143 (2021).
63. Zhao, Z. et al. Redox-active azulene-based 2D conjugated covalent organic framework for organic memristors. *Angew. Chem. Int. Ed.* **62**, e202217249 (2023).
64. Tao, Y. et al. Electrochemical preparation of porous organic polymer films for high-performance memristors. *Angew. Chem. Int. Ed.* **61**, e202209952 (2022).
65. Wang, L. et al. Viologen-hypercrosslinked ionic porous polymer films as active layers for electronic and energy storage devices. *Adv. Mater. Interfaces* **5**, 1701679 (2018).
66. Liu, L. et al. A highly crystalline single layer 2D polymer for low variability and excellent scalability molecular memristors. *Adv. Mater.* **35**, 2208377 (2023).
67. Yang, X. et al. Solution-processed hydrogen-bonded organic framework nanofilms for high-performance resistive memory devices. *Adv. Mater.* <https://doi.org/10.1002/adma.202305344> (2023).

REVIEWERS' COMMENTS

Reviewer #1 (Remarks to the Author):

The authors have addressed my comments carefully and correctly. It can be accepted now

Reviewer #2 (Remarks to the Author):

The authors have addressed all the comments and improved the manuscript apparently, I recommend acceptance for publication as is.

Responses to Reviewers' Comments

Manuscript number: NCOMMS-23-27012A

Title: Multistate structures in a hydrogen-bonded polycatenation HOF with diverse resistive switching behaviors

REVIEWERS' COMMENTS

Reviewer #1 (Remarks to the Author):

The authors have addressed my comments carefully and correctly. It can be accepted now.

Response: We thank Reviewer #1 for their positive remarks.

Reviewer #2 (Remarks to the Author):

The authors have addressed all the comments and improved the manuscript apparently, I recommend acceptance for publication as is.

Response: We thank Reviewer #2 for their positive remarks.